# Agricultural Economic Growth, Renewable Energy Supply and CO$_2$ Emissions Nexus

**Tagwi Aluwani** 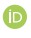

Department of Agriculture and Animal Health, School of Agriculture and Life Sciences, University of South Africa, Johannesburg 1709, South Africa; tagwia@unisa.ac.za

**Abstract:** International trade has created more economic growth opportunities in the agriculture sector. The agricultural sector remains key to the South African economy, with a vibrant international market becoming available as the country's agriculture exports grow. However, the impacts of human-caused global warming have intensified as a result of increased greenhouse gas emissions, notably carbon dioxide (CO$_2$), which negatively affects agricultural productivity and the economy. Considering the future energy resource demands for agricultural productivity due to the expected population growth and the emphasis on environmental remedial actions, the following question presents itself: what impact will a clean energy supply have on the agricultural economy and the environment, notwithstanding that agriculture, as a sector, also has a huge potential to contribute to renewable energy production? This study examines the effect of the nexus of South Africa's renewable energy supply, CO$_2$ emissions and trade openness on agricultural economic growth from 1990 to 2021. The nexus provides crucial insights into policies targeted at promoting renewable energy in the agricultural sector by isolating key areas of priority. An autoregressive distributed lag (ARDL) bounds test, fully modified ordinary least square (FMOLS) test, a dynamic ordinary least square (DOLS) test and a canonical cointegrating regression (CCR) econometric analysis were used to estimate the nexus. The results showed that growth in the agricultural sector leads to deterioration in the environment, while international trade benefits the sector. The scale of renewable energy supply slowed down the agricultural economy. The study makes a new contribution in providing empirical evidence for the links between renewable energy supply and agricultural GDP, which can drive policy on renewable energy use in the agricultural sector in South Africa. The paper recommends intentional renewable energy production research and development (R&D) finance focusing on renewable energy human development planning and investments in vocational programmes in higher learning institutes, agricultural renewable energy policy and the creation of green incentive schemes for feedstock producers, especially in rural areas in the agricultural sector.

**Keywords:** agricultural economic growth; renewable energy; economic growth; ARDL; South Africa; error correction model

## 1. Introduction

More than ever before, agricultural economic growth is crucial in the context of international trade and self-sustenance. The recent COVID-19 pandemic has fully demonstrated this fact. Energy is an important component of economic growth, even more so in the agricultural sector. The major challenge at hand is that economic growth has a negative relationship with environmental quality, while the world economy heavily depends on fossil energy to function. With the expected rise in demand for food, especially due to the rise in population, the agricultural sector faces more pressure to feed the world, at the same time reducing its carbon footprint. The energy sector and agriculture constitute the largest share of global pollution. Greenhouse gas emissions (GHGs) from the agricultural sector, such as carbon dioxide (CO$_2$), methane (CH$_4$) and nitrous oxide (N$_2$O), contribute to climate change and, at the same time, the sector is acutely affected by its effects (Holka

et al. 2022). The global pressure to maintain global average temperatures between 1.5 and 2 degrees Celsius pre-industrial level is a priority for the world. Although studies have shown that renewable energy use can offset some of the adverse consequences of fossil fuels, most economies cannot afford low-carbon energy technologies, especially in Africa; this is an indication that there is a need for a multifaceted approach when solving climate change issues. The reality is that poorer nations' lack of climate finance perpetuates climate injustice. Lamb et al. (2021) posit that if low-carbon technologies and practices are allowed to progressively phase in on an unfair playing field, climate mitigation cannot be achieved. In addition, the lack of sanctions creates more challenges as it may deter nations from compromising environmental protection in favor of economic growth, especially in emerging economies where financial resources are few (Caetano et al. 2022). This is a recognition that it will take more financial commitment for regions such as Africa to transition from fossil to clean energy.

In South Africa, the economy is coal driven, with more than 70% of coal energy generation. The uptake of renewable energy is still in its infancy. However, there are many opportunities for the agricultural sector to reduce GHG emissions by using and producing renewable energy. The agricultural sector has a wealth of organic feedstock that can be used to produce bioenergy, which can increase carbon sequestration (Khan et al. 2022b). The status quo is that the renewable energy sector still suffers from a small market share worldwide in general (Al Yousif and Yousif 2020), which inflates prices and causes low uptake. Nevertheless, in South Africa, renewable energy presents a myriad of opportunities for farmers who can participate as producers as well as contribute to emissions reduction as consumers of farming activities (Tagwi and Chipfupa 2022). Rapid industrialization also presents more opportunities for the demand and supply of renewable energy on the African continent, which will be driven by increased agricultural activities. Currently, the South African agricultural sector is highly export orientated, with major destinations being Africa, Asia and Europe (Sihlobo 2022). This trajectory indicates an economy that is open and expanding. Considering the political and economic historical imbalances, the economy must grow to improve the per capita welfare of the country. Of note, a looming challenge for developing countries is their inevitable rapidly rising populations and incomes which put immense pressure on natural resources, consequently increasing agricultural emissions, especially in sub-Saharan Africa (Searchinger et al. 2018; Ali et al. 2019a; Sands and Suttles 2022). Agricultural land use, deforestation and other land use changes contribute to carbon emissions. Emissions from livestock constitute most of the pollution across the world in the agricultural sector (Lamb et al. 2021; Dumas et al. 2022). This has implications for food and livestock demand, which automatically exerts pressure on the production side. However, the agricultural sector has a lot of organic feedstocks that can be used as biodigestate for the production of renewable energy (Ali et al. 2022). This, then, qualifies the sector to take more carbon footprint reduction measures. Moreover, within the continent, South Africa emits almost half of the emissions, as depicted in Figure 1 (OWID 2022). On the continent, the SADC region is expected to be the biggest emitter in the agricultural sector due to its expected large livestock operations in the future (Seketeme et al. 2022). Notwithstanding that the development of economies has been the main driver of carbon emissions (Li and Wei 2021; Kongkuah et al. 2022), and that the agricultural industry has equally played a part, which has caused climate change and climate variability, renewable energy can benefit the sector. The side effects of climate change have reached unprecedented levels and action can no longer be postponed (Hugonnet et al. 2021). The agricultural sector has been the hardest hit by climate change effects (Ait-El-Mokhtar et al. 2022). From the literature, it is evident that economic growth and the environment, energy and trade are closely associated. However, South Africa still experiences many challenges, including an energy crisis, and with the country ranking first among 164 countries as the most unequal country in the world (World Bank 2022), economic growth will be a top priority for years to come. However, the renewable energy transition will require massive green technology infrastructure investments. Tendengu et al. (2022) found that government expenditure

increase that is not capital in nature has not been benefiting the South African economy from 1988 to 2018, and they advocate for infrastructure investment prioritization as a remedial action. Infrastructure facilitates agricultural production activities which increases agricultural growth (Boni 2022). Investments in green technology infrastructure in the agricultural sector should be prioritized. It is now clear that the agricultural sector will be subjected to immense pressure due to the rising population. It is also a fact that the current trend of global warming is catastrophic and the need to transition to renewable energy is undeniable. On the other hand, the trajectory of fossil energy production costs is also upward and not sustainable; therefore, transitioning to green technologies should be an incentive for developing economies. Agricultural sector growth is a necessity as it is the major employer of poor households, mostly situated in rural areas. In the context of sustainable agricultural economic growth, the paper aims to gain an understanding of the extent to which clean energy supply and the environment affect the agricultural economy in an open-market environment. The paper seeks to answer the following questions: What is the impact of agricultural economic growth on the environment? What is the nature of the relationship between international trade and agricultural economic growth? Is the scale of renewable energy supply improving the agricultural economy? The interest in the production side of renewable energy is primarily because the agricultural sector can play a major role in the supply side as feedstock for bioenergy comes from the sector. The paper hypothesizes environmental degradation as the economy grows, as informed by the environmental Kuznets Curve hypothesis. Although trade openness has been found to affect economies positively and negatively (Chen et al. 2022b), on the bright side, the benefits are associated with improvement in technology, human capital, infrastructure, innovation and competitiveness. However, these improvements are subject to the countries' developmental stages (Darku and Yeboah 2018). The study hypothesizes improvement of the agricultural economy as a result of trade based on the general sectorial economic improvement that South Africa has made since democracy. A priori, the renewable energy supply was expected to play an insignificant role in the economy due to the low scale of consumption, which also reflects the potential scale of consumption. Based on these insights, the study's interest is on the supply side rather than the consumption side precisely because renewable energy supply also indicates the capacity of the country to meet local green-energy demands without the effect of importation.

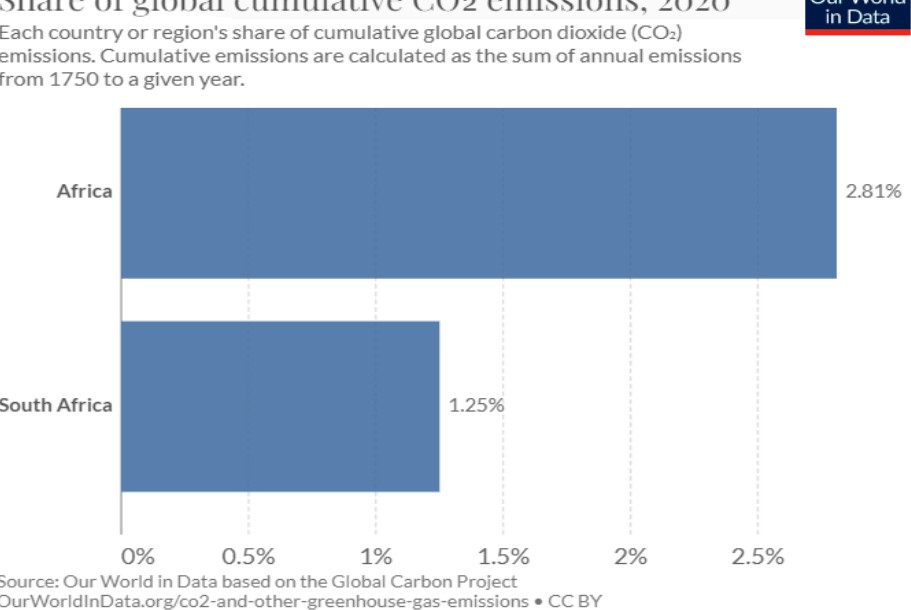

**Figure 1.** South Africa's share of emissions in Africa.

The paper makes the first contribution to the discourse of renewable energy production, specifically in the agricultural sector in South Africa. Other researchers have studied the relationship between agricultural economic growth and the environment with the consideration of renewable energy consumption. These researchers include Liu et al. (2017) who focused on real GDP, fossil energy consumption, renewable energy consumption, $CO_2$ and agricultural value-added (net output) variables. Usman and Makhdum (2021) used financial development, ecological footprint, forest area, fossil energy consumption, renewable energy consumption, $CO_2$ and agriculture value-added (per worker) variables. Shah et al. (2022a) used ICT, human capital index, GDP, renewable energy consumption, $CO_2$ and agriculture value-added variables (%GDP). Shah et al. (2022b) considered total patent application, environmental policy stringency index, revised combined polity score, fine particulate matter with a diameter of 2.5 μm or less, ICT, renewable energy consumption, $CO_2$ and agriculture value added variables (per worker). However, all these studies focused exclusively on the BRICS bloc and not specifically on South Africa. The studies modelled various variables and only focused on renewable consumption and not supply. In light of these insights, there is a gap, and the study seeks to close the gap by investigating the South African agricultural economy's relationship with renewable energy production, the environment and trade openness. The remaining sections of the paper are structured as follows: Section 2 covers the theoretical framework, while review of the literature is under Section 3. The methodology is presented in Section 4. The empirical results and discussion are presented in Sections 5 and 6, respectively. Conclusions and policy recommendations are presented in Section 7.

## 2. Theoretical Framework

The discourse around economic growth and environmental quality is that the accumulation of production factors, which raises firms' need for polluting inputs, is the primary cause of income growth (Lopez 1994, 2017). Economic growth and environmental quality have been associated with a positive and negative relationship within the broader Environmental Kuznets Curve (EKC) hypothesis (Kuznets 1955). According to the hypothesis, within the short term, the expected outcome is positive, but it is negative in the long run (Grossman and Krueger 1991). The positive effect represents the scale effect, while the negative relationship represents the technique effect (Udeagha and Muchapondwa 2022). This implies that as the agricultural economy grows, the environment will also be deteriorating in the short run; however, as income increases, production methods will move away from heavily industrialized methods which emit more emissions and become more service orientated (emit fewer emissions). However, this is conducted at the expense of poorer economies. According to the "pollution haven" hypothesis, wealthy nations export polluting industrial activities to poorer nations where environmental rules are relaxed (Bardi and Hfaiedh 2021). This happens primarily because as income rises, citizens prioritize the quality of life and exert more pressure on the government to implement environmentally friendly policies. Bashir et al. (2022, 2020a, 2020b) also observed that companies with relaxed green technologies also migrate to poorer countries, affecting the environment quality negatively. The EKC has three stages, namely, the uptrend stage with low income (pre-industrial), driven by economic inefficiencies, the mass production stage (industrial), with the rise in income, and lastly, the green stage (post-industrial), with a rise in income but with more green technologies (Dinda 2004). The subject of trade liberalization has gained traction over the years and has proven to be key to economic growth. Since the early 1980s, most emerging economies have undertaken economic liberalization initiatives driven by the debt crisis. The outward-looking policies adopted by Asian economies have enhanced trade openness and contributed to GDP growth (Çevik et al. 2019). Chen et al. (2019) posits that renewable energy, trade and growth form the nexus as an ecosystem with a snowball effect. Thus, trade can induce renewable energy production, which can stimulate renewable energy consumption, which can further stimulate more production of renewable energy. Among other things, supply, demand, imports and exports frequently

have an impact on the energy markets (Marques et al. 2019). Considering that renewable energy production is cleaner, if its growth increases with agricultural economic growth, such growth is desirable for the environment. Based on this framework, the study seeks to assess the nature of the relationship between agricultural economy, renewable energy production as a stimulant of consumption, the environment and trade.

## 3. Literature Review

### 3.1. Economic Growth and the Environment

The sustainable economic growth agenda has placed environmental pollution at the centre stage. Economic agents are now forced to reimagine the way energy is produced. It is a well-established fact that economic development exerts pressure on natural resources which reduces environmental quality. However, there is no consensus on the impact across the globe. Different studies have shown positive and negative relationships between economic growth and environmental quality. Agricultural production in BRICS countries was also found to decrease environmental quality. The study included renewable energy supply, ICT and GDP variables. The Augmented Mean Group (AMG) estimator method was used for data analysis ranging from 1990–2019 (Shah et al. 2022a). Using the ARDL approach during the 1961–2018 period, Kılavuz and Doğan (2021) found that economic growth deteriorated the environment in Turkey. In Bashir et al. (2022), a study using the method of moment quantile regression (MMQR), AMG and CCEMG analysis for the newly industrialized countries (NIC) between 1990 and 2018 revealed that economic growth was among the variables contributing to environmental deterioration. In Tunisia, Mbarek et al. (2018) found that economic growth increased carbon emissions using a Vector error correction model (VECM). Other variables used in the study included renewable energy consumption, and fossil energy consumption during the 1990–2015 period. Using system-GMM and quantile regression, Bashir et al. (2020c) also observed a decrease in environmental quality as the economy grew in OECD economies from 1995 to 2015. The study considered other variables such as renewable energy consumption, environmental taxes, environmental technology and financial development. In Malaysia between 1978 and 2016, using the ARDL approach, carbon emissions were found to have significantly increased due to economic growth (Ridzuan et al. 2020). Using ARDL and the dynamic ordinary least squares (DOLS) method in Peru, Raihan and Tuspekova (2022e) observed that economic growth increased agricultural land expansion from 1990 to 2018. Using the wavelet coherence method and ARDL technique in South Africa between 1971 and 2016, Adebayo and Odugbesan (2021) found a positive relationship between carbon emission and economic growth. Similar results were observed in Japan using wavelength tools from 1990Q1 to 2015Q4 (Adebayo and Kirikkaleli 2021). Using earlier data from 1960 to 2010 for Japan, Rahman et al. (2022) also found a positive relationship between economic growth and environmental degradation using the ARDL approach. In Indonesia, Massagony and Budiono (2022) also found a linear relationship between economic growth and the environment using the ARDL technique. Agboola et al. (2022) also found economic expansion reduced environmental quality in the scale stage from 1970 to 2020 using the dynamic autoregressive-distributed lag. These findings were consistent with most research outcomes on the environment and economic growth. Studies have shown that economic growth and environmental nexus studies are inconclusive. Using FMOLS and DOLS in Malaysia during the period 1970–2009, Begum et al. (2015) found an inverse relationship between economic growth and environmental quality, implying that economic growth improved the environment. Other studies have also observed a positive relationship between economic growth and environmental degradation (Nihayah et al. 2022).

### 3.2. Economic Growth and Trade

The outward-looking approach to trade has been hailed as a growth generator with evidence from East Asian countries. With the advent of trade liberalization, countries have seen economic growth. Trade liberalization is the relaxation of tariff and non-tariff barriers

that limit trade (Muhammed et al. 2022; Mignamissi and Nguekeng 2022). Zafar et al. (2019) argued that trade can either improve or reduce environmental quality, depending on the production technologies employed. Using the structural equation modelling method for Southeast Asian and Latin American countries from 1991 to 2018, an increase in trade was found to improve economic growth (Zeeshan et al. 2022). According to Afolabi (2022), trade openness has various advantages, such as foreign direct investment, technology transfer, transfer of goods, services and transfer of capital. Shrestha's (2022) study, which focused on analyzing the impact of trade liberalization policies, found that trade openness was decreasing agricultural growth in Nepal and increasing dependency on foreign agricultural goods. Wani (2022), using 1993–2019 data, and SenGupta (2020), using 1960–2018 data, also discovered a long-term and short-term inverse relationship between trade openness and economic growth in India, implying that trade had negative effects on economic growth, while heavy market dependency on foreign markets could also have played a role. Similar findings were observed by Malefane and Odhiambo (2021) in the case of Lesotho, using data from 1979 to 2013, using the ARDL approach. The different outcomes confirm that trade openness is a double-edged sword; it can build resilience and also create vulnerability, and this suggest that the state and the priorities of the countries' development will determine the extent to which countries will capitalize on the open markets.

*3.3. Agricultural Economic Growth and Renewable Energy Supply*

The basic building blocks of the world economy are coal, natural gas and crude oil, and the cost of oil production has increased. By increasing product prices to reflect increased energy and raw material costs, the economy has made the necessary adjustments. The problem is that even though renewable energy sources have long been acknowledged in theory in economic and social practice, they are still viewed as a problem for the future (Kircher 2019). Of concern, despite a series of decarbonization policy discussions for years, is that consensus on achieving this goal has not been reached (Khabbazan and Hokamp 2022). Developed countries have embraced the use of renewable energy in agriculture; however, poor nations are still having difficulty in the implementation due to technical and economic challenges (Rahman et al. 2022). The discourse of renewable energy as a mitigator has gained traction across the continent due to the rapid increase in emissions. In a review study by Lamb et al. (2021), looking at the trends and drivers of greenhouse gas emissions by sectors from 1990 to 2018, agricultural activities were found to be increasing emissions in Africa and the trend trajectory is upward. This observation indicates that it is therefore crucial for countries to be geared towards renewable energy investments. A study by Banks and Schäffler (2005) concluded that increased usage of renewable energy would also lessen South Africa's reliance on the fluctuating (and rising) costs of imported fuels. Looking at the current price fluctuations of fossil fuel due to geopolitical challenges, the volatility will be difficult to manage in future in the presence of other shocks, and the poorest countries will experience the full brunt. The negative spin-offs will be felt in the agricultural sector currently dominated by small-scale farmers with fewer resources. Akinbami et al. (2021) also looked at the state of renewable energy development in South Africa and concluded that South Africa is endowed with enormous biomass, wind and solar energy potential, and waste management systems and investments should be given attention. The current challenge is that focus has been directed to wind and solar investments while modern biomass potential is untapped. Recent investments in South Africa in renewable energy have been in solar and wind through the Renewable Independent Power Producer Programme (REIPPP). The biomass potential resides within the agricultural sector as farmers are mostly the producers of this biomass. If this opportunity is explored, the agricultural sector stands to benefit as farmers can produce energy, ultimately boosting the scale of production. This can reduce energy costs for farmers and excess energy can also be sold to the national grid. In addition, Uhunamure and Shale (2021) looked at the SWOT analysis of renewable energy generation in South Africa, and concluded that geographic position, political and economic stability and policy implementation were

South Africa's strengths, while government bureaucratic processes, level of awareness and high investment costs formed part of its weaknesses. The lack of awareness identified in the SWOT analysis highlights the need for an aggressive approach by the government to publicize renewable energy at all levels, but with a special focus on farmers and businesses located in rural areas to ensure that no one is left behind as per just energy transition goals. Over time, this will ensure a seamless adoption of renewable energy technologies. Ibrahim et al. (2021) reviewed African renewable energy production using Nigeria, Cameroon, Ghana and South Africa as a case study. The study recommended tax rebates on renewable energy to encourage energy production. Tax subsidies for renewable energy use are important; however, because the problem at hand is largely that of low supply which inflates prices and discourages adoption, the South African government should focus on incentivising producers more. In the agricultural sector, this initiative could reduce farming areas' reliance on the main grid and also build sustainable ecosystems in rural areas where most of the farms and biomass are situated. The economic ripple effect benefit could be huge in the future. Aliyu et al. (2018) reviewed renewable energy development in Africa, focusing on South Africa, Egypt and Nigeria, and recommended a focus on technology, awareness and skills development for renewable energy production. Considering that renewable energy is a relatively new economy, skills shortages are a major problem and will require a bottom-up approach in South Africa. This will require renewable energy curriculum development, something currently not existing in vocational institutions of higher learning. This effort can then merge well with technology training. Such efforts can fast-track technology transfer in the energy economy. For an effective clean energy policy development, it is imperative to understand the dynamics and linkages involved between the agricultural sector and trade, clean energy supply and the environment. In South Africa, a gap still exists and, to date, few to no studies have modelled these dynamics.

Available studies modelling similar relationships between the agricultural economy, renewable energy, the environment and trade include a study by Chandio et al. (2021), which examined the effects of China's economic growth, agricultural productivity, renewable energy consumption and forestry area on $CO_2$ emissions from 1990 to 2015 using ARDL and the fully modified ordinary least square (FMOLS) method. However, the study used renewable energy consumption and did not include renewable energy supply and trade openness. In Okumus et al. (2021), the association between renewable and non-renewable energy consumption and economic growth was estimated in G7 countries from 1980–2016, using the CS-ARDL method. The results predicted that renewable energy increases economic growth. Renewable energy supply and trade openness were excluded from the model. In Iran, a similar study was conducted looking at the linkages between renewable energy use, carbon emissions and economic growth from 1975–2017 using a non-ARDL model. The results predicted that renewable energy increased economic growth (Karimi et al. 2021). The study did not include trade and renewable energy supply. Busu (2020) also analyzed the impact of renewable energy sources on economic growth in the EU using the ARDL technique. The results predicted that renewable energy increased economic growth. Magazzino et al. (2022) also assessed renewable energy consumption, environmental degradation and economic growth nexus from 1990–2018 in Scandinavian countries using an FMOLS technique. The results predicted a positive relationship between renewable energy and economic growth. A study by Pata (2021), using the Fourier ADL test to examine the effects of agricultural practices, globalization, and renewable energy production on ecological footprints and carbon dioxide ($CO_2$) emissions in BRIC countries for the period of 1971–2016, found that globalization increased pollution, while renewable energy improved the environment in Brazil. The study's conclusions reaffirmed the value of renewable energy. The studies showed that renewable energy consumption improved the economy. However, in similar studies, contrary results were observed from Akram et al. (2021), Fotio et al. (2022) and Ozturk et al. (2022), using PQR, PMG–ARDL model and PVAR, respectively, which concluded that renewable energy decreased economic growth. From the literature, it is clear that renewable energy, economic growth and carbon emission nexus elicit different views.

Of note, all the studies did not include trade openness and renewable energy supply in the modelling, and this is what the current study seeks to explore. Moreover, the contribution of renewable energy to economic expansion cannot be understated. The generation of bioenergy can greatly benefit the agricultural industry. According to Duque-Acevedo et al. (2020), the agricultural sector generates a lot of biomass waste and to explore prospects, the authors recommended that governments implement biomass waste management models and make investments in R&D. The observed positive and negative relationships between economic growth, $CO_2$ emissions, renewable energy and trade openness indicate that the impact varies widely across countries; therefore, countries should take an individual approach in dealing with the energy–growth–environment nexus.

## 4. Methodology

Due to the unavailability of data, the study's data period was chosen to be from 1990 to 2021 for all variables. Table 1 shows all the variables used in the study. Variables used included annual agricultural GDP in 2015 constant in USD and trade openness ($\sum$ exports, imports) in 2015 constant in USD. Data was sourced from the World Bank Development Indicators (WDI). Carbon dioxide emissions ($CO_2$) annual data were measured in million tonnes (Mt), and renewable energy production (including hydroelectricity, nuclear, solar, wind, geothermal, biomass and other), measured in terawatt-hours (TWh), was sourced from BP statistics. All variables were transformed into logarithms for elasticity reporting. The descriptive statistics for the variables utilized in this study are listed in Table 2. ARDL modelling was used, a technique for examining cointegrating relationships when dealing with the combination of I(0) and I(1) variables, which are variables that are stationary at a certain level and those that are stationary only after first differencing. In recent years, this technique has attracted renewed interest. Credit for this technique goes to Pesaran et al. (2001). The model presents a balance testing approach for examining cointegration and is divided into two components: the short run and long run. This is a desirable trait and allows conclusions to be drawn about the short-run and long-run effects. ARDL is preferred for dynamic estimation, despite some regressors' endogeneity, as the ARDL technique offers accurate t-statistics and unbiased estimates and is intuitively desirable for dynamic estimation (Harris and Sollis 2003; Jalil and Ma 2008). The optimal lag selection also eliminates residual correlation correcting for endogeneity (Ali et al. 2016). Moreover, the ARDL model choice was preferred due to its ability to make inferences that can guide policy in the short and long run. The model can estimate areas of intervention in achieving sustainable agricultural economic growth for policy makers. The model was also used by various authors in agriculture, economic growth, renewable energy, and environment nexus (Ali et al. 2019b; Aziz et al. 2020; Yurtkuran 2021; Usman et al. 2022; Wang 2022). In addition, the ARDL model is suitable for small-sample data estimation. The model has many advantages (flexibility, interpretability, eloquence and statistical properties) (Menegaki 2019). FMOLS, DOLS and CCR model techniques were also used to confirm the robustness of the model. The disadvantage of ARDL is that these two components introduce complexity to the model and a long-run relationship must be established before the error correction model is estimated. Thus, even if the series is stationary, in the absence of long-term cointegration, long-term inferences cannot be made. The multivariate sample model is expressed in Equation (1).

$$Y_t = \alpha_0 + \beta_1 X_1 + \beta_2 X_2 + \beta_3 X_3 + \varepsilon_t \tag{1}$$

The relationship can be expressed as follows:

$$LnAgric\_GDP = f(LnRENP, \ LnCO_2, \ LnTRADE\_OPEN) \tag{2}$$

where *LnAgric_GDP* stands for agricultural gross domestic product, *LnRENP* is renewable energy production, *LnCO_2* is carbon dioxide emissions and lntrade_open is trade openness.

**Table 1.** Descriptive statistics.

| Variable | Description | Source |
|---|---|---|
| $LNCO_2$ | Carbon Emissions | BP |
| LNRENP | Renewable energy production | BP |
| LNAGRIC_GDP | Agricultural gross domestic value added (constant 2015 in USD) | WDI |
| LNTRADE_OPEN | ($\sum$ exports, imports) (constant 2015 USD) | WDI |

**Table 2.** Descriptive statistics.

|  | LNAGRIC_GDP | $LNCO_2$ | LNRENP | LNTRADE_OPEN |
|---|---|---|---|---|
| Mean | 22.54503 | 6.011119 | 2.735773 | 2.746766 |
| Median | 22.46673 | 6.073114 | 2.686812 | 2.697060 |
| Maximum | 23.03748 | 6.165323 | 3.413082 | 3.427212 |
| Minimum | 22.07652 | 5.765885 | 2.001615 | 2.011787 |
| Std. Dev. | 0.235645 | 0.138572 | 0.335024 | 0.336262 |
| Skewness | 0.260765 | −0.401812 | 0.416847 | 0.424582 |
| Kurtosis | 2.386492 | 1.622783 | 2.977247 | 2.976266 |
| Jarque-Bera | 0.864514 | 3.390052 | 0.927416 | 0.962188 |
| Probability | 0.649042 | 0.183594 | 0.628947 | 0.618107 |
| Sum | 721.4411 | 192.3558 | 87.54472 | 87.89652 |
| Sum Sq. Dev. | 1.721389 | 0.595266 | 3.479472 | 3.505239 |

The equation can be further fitted as follows:

$$LnAgric\_GDP = \alpha_0 + \beta_1 LnRENP_t + \beta_2 LnCO_{2_t} + \beta_3 LnTRADE\_OPEN_t + \varepsilon_t. \quad (3)$$

The *ARDL* model is specified in Equation (4) where $\gamma_1$ and $\varnothing_1$ capture long- and short-run elasticities coefficients, while *p* and *q* denote the lag length for the regress and regressors, respectively, and $\varepsilon_t$ is the white noise disturbance term.

$$
\begin{aligned}
\Delta LnAgric\_GDP_t &= \alpha_0 + \gamma_1(LnRENP)_{t-1} + \gamma_2(LnCO_2)_{t-1} + \gamma_3(LnTRADE\_OPEN)_{t-1} \\
&+ \sum_{i=1}^{p} \varnothing_1\Delta(LnAgric\_GDP)_{t-1} + \sum_{i=1}^{q} \varnothing_2\Delta(LnRENP)_{t-1} + \sum_{i=1}^{q} \varnothing_3\Delta(LnCO_2)_{t-1} \\
&+ \sum_{i=1}^{q} \varnothing_4\Delta(LnTRADE\_OPEN)_{t-1} + \varepsilon_t
\end{aligned} \quad (4)
$$

The first component of the equation represents the long run and is specified as follows:

$$
\Delta LnAgric\_GDP_t = \alpha_0 + \sum_{i=1}^{p} \gamma_1(LnAgric\_GDP)_{t-1} + \sum_{i=1}^{q} \gamma_2(LnRENP)_{t-1} + \sum_{i=1}^{q} \gamma_3(LnCO_2)_{t-1} + \\ \sum_{i=1}^{q} \gamma_4\Delta(LnTRADE\_OPEN)_{t-1} + \varepsilon_t. \quad (5)
$$

The error correction model (ECM) tests for long- and short-run causality of the series in the model. The ECM component determines the causation of the series in at least one direction.

The equation below is specified as follows:

$$
\Delta LnAgric\_GDP_t = \alpha_0 + \gamma_1(LnRENP)_{t-1} + \gamma_2(LnCO_2)_{t-1} + \gamma_3(LnTRADE\_OPEN)_{t-1} + \\ \sum_{i=1}^{p} \varnothing_1\Delta(LnAgric\_GDP)_{t-1} + \sum_{i=1}^{q} \varnothing_2\Delta(LnRENP)_{t-1} + \sum_{i=1}^{q} \varnothing_3\Delta(LnCO_2)_{t-1} + \\ \sum_{i=1}^{q} \varnothing_4\Delta(LnTRADE\_OPEN)_{t-1} + \varepsilon_t; \quad (6)
$$

$$
\Delta LnAgric\_GDP_t = \alpha_0 + \gamma_1(LnRENP)_{t-1} + \gamma_2(LnCO_2)_{t-1} + \gamma_3(LnTRADE\_OPEN)_{t-1} + \\ \sum_{i=1}^{p} \varnothing_1\Delta(LnAgric\_GDP)_{t-1} + \sum_{i=1}^{q} \varnothing_2\Delta(LnRENP)_{t-1} + \sum_{i=1}^{q} \varnothing_3\Delta(LnCO_2)_{t-1} + \\ \sum_{i=1}^{q} \varnothing_4\Delta(LnTRADE\_OPEN)_{t-1} + \delta ECM_{t-1} + \varepsilon_t. \quad (7)
$$

ECM is the speed of adjustment from the long-run disequilibrium to the short-run equilibrium and it corrects the disequilibrium. The component indicates the speed of convergence to the equilibrium in the presence of shocks. The $\delta ECM$ is expected to be significantly negative, with a value less than or equal to 1. The existence of cointegration suggests a long-run relationship in the model.

## 5. Results

### 5.1. Descriptive Statistics

The descriptive statistics of all variables in this study are shown in Table 2, in which the mean value of the dependent variable LNAGRIC_GDP is 22.5 and the standard deviation is 0.24. The mean values of explanatory variables, $LNCO_2$, LNRENP and LNTRADE_OPEN, were 6.01, 2.75 and 2.75, respectively, and the standard deviations were 0.13, 0.34 and 0.34, respectively. The behavior of renewable energy production and trade openness was volatile, showing more variation when compared to agricultural GDP and $CO_2$ emissions. The kurtosis value for all variables was less than 3 while the JB test for normality was more than 1% indicating normal distribution for all variables. The historical trend of the variables is displayed in Figure 2a–d.

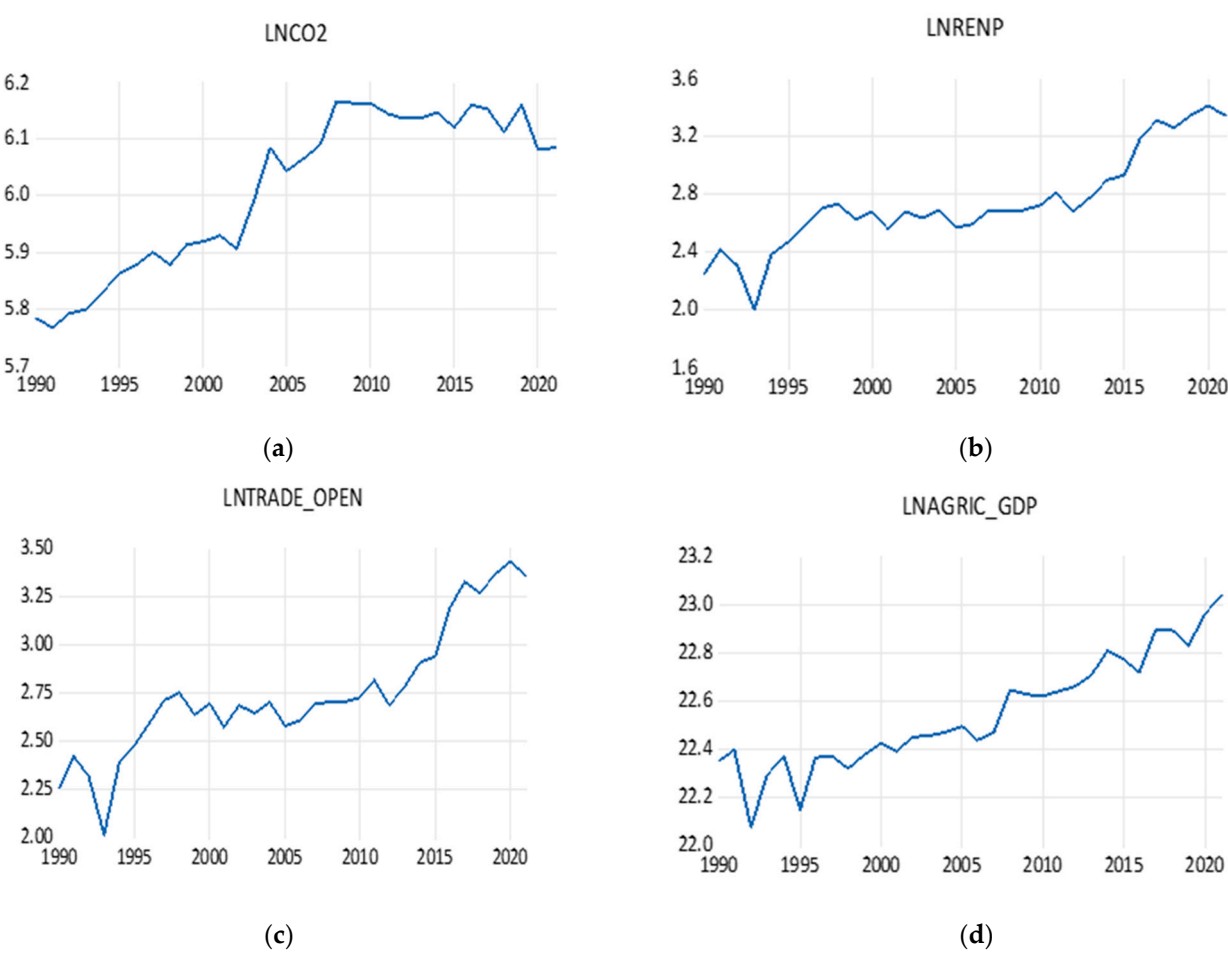

**Figure 2.** (**a**) Carbon emissions trend; (**b**) renewable energy supply trend; (**c**) trade openness trend; (**d**) agricultural growth trend.

### 5.2. Coefficient Correlation

The results in Table 3 showed a stronger correlation between agricultural gross domestic value and carbon emissions (0.79), renewable energy production (0.87) and trade openness (0.87). There is also a stronger correlation between trade openness and carbon emissions (0.70), renewable energy production (0.99) and agricultural GDP (0.87). The higher

correlation values demonstrate that most variables have a strong link with one another because of their shared characteristics (macroeconomic variables) (Tabash et al. 2022).

**Table 3.** Correlation Matrix.

|  | LNCO$_2$ | LNRENP | LNTRADE_OPEN | LNAGRIC_GDP |
|---|---|---|---|---|
| LNCO$_2$ | 1 | 0.6986692684970441 | 0.6977271969555639 | 0.7925232751831976 |
| LNRENP | 0.6986692684970441 | 1 | 0.9999973314476189 | 0.8694416348428991 |
| LNTRADE_OPEN | 0.6977271969555639 | 0.9999973314476189 | 1 | 0.8695073092644545 |
| LNAGRIC_GDP | 0.7925232751831976 | 0.8694416348428991 | 0.8695073092644545 | 1 |

*5.3. Unit Root Test*

In time series, the problem of non-stationarity must be addressed, failure of which results in spurious regression (Nelson and Plosser 1982). Variables with a unit root give rise to unreliable interpretations. To correct this problem of explosiveness or seasonal unit root, a unit root test must be conducted to ascertain if there is no series integrated of order 2 or higher. If the results conclude that there is an absence of unit root in the first differences—that is, if series are either of order (0) and order I(1)—we can proceed to the second step. The results in Table 4 show that the series were stationary at level I(0) and the first difference I(1). The ARDL method is thus appropriate for the analysis due to the level of stationarity that exists in a mixture of order levels I(0) and I(1) (Jordan and Philips 2018). The VAR and VECM models can then be used in the analysis. Various tests such as Dickey–Fuller (DF-GLS), Augmented Dickey–Fuller (ADF), Phillips–Perron (PP), Kwiatkowski–Phillips–Schmidt–Shin (KPSS), Narayan and Popp and the second-generation unit root test suitable for data with structural breaks by Zivot and Andrews (Zivot and Andrews 2002) are used for unit root analysis; however, PP and ADF were used. Table 4 shows the outcomes of the two-unit root tests, Augmented Dickey–Fuller (ADF) (Dickey and Fuller 1979) and Phillips–Perron (PP) (Phillips and Perron 1988). The findings of the ADF and PP unit root tests are comparable. Although traditional stationarity tests do not account for structural breaks, there was no evidence of structural breaks in the data. Stationarity is achieved in series when the variance, covariance and mean are constant (Khan et al. 2022a). ADF results show that all variables are integrated at I(1), while PP results show that only agricultural GDP is integrated at I(0) and the rest of the variables at I(1). Therefore, the results indicate that the null hypothesis of a unit root is rejected, and all variables are not of order I(2).

ADF general equation is specified as follows:

$$\Delta x_t = \delta x_{t-1} + \sum_{i=1}^{m} \varphi \Delta x_{t-1} + \varepsilon_t.$$

*5.4. Cointegration Test*

The cointegration test determines whether a long-term equilibrium relationship exists between endogenous and exogenous variables. Traditional cointegration test methods such as the Engle and Granger (1987) test, Johansen (1988) and Johansen and Juselius (1990) are used strictly on variables integrated of the same order. However, Pesaran et al. (2001) created an ARDL-bound testing technique solution for the cointegration of variables of different orders. Thus, if the variables have different ordering and some of them are I(0) stationary while others are I(1) stationary, the cointegration test can still be run. The bound approach estimates an unrestricted conditional error correction model where each variable is accounted for as a dependent variable. The bound test generates two sets of critical values: lower- and upper-bound values. If the F value is below the lower bound, the null hypothesis of no cointegration fails to be rejected. A long cointegration relationship exists if the F statistic value generated by the bounds testing is larger than the upper bound I(1), and the null hypothesis is rejected. An F value that falls between lower and upper suggests

that the cointegration relationship is inconclusive and therefore inferences cannot be made. In Table 5, ARDL bounds testing results are shown.

**Table 4.** Unit root test analysis.

| Series | Model | ADF | ADF-P | PP | PP-P |
|---|---|---|---|---|---|
| At Level—I(0) | | $\tau_\mu \ \tau_\tau \ \tau$ | Value | $\tau_\mu \ \tau_\tau \ \tau$ | Value |
| LNCO$_2$ | Intercept ($t_m$) | −1.6075 | 0.4669 | −1.6615 | 0.4402 |
| | Intercept and Trend ($t_t$) | −0.8719 | 0.9468 | −0.6463 | 0.9685 |
| | None (t) | 1.4163 | 0.9577 | 1.4163 | 0.9577 |
| LNRENP | Intercept ($t_m$) | −0.7936 | 0.807 | −0.6881 | 0.8354 |
| | Intercept and Trend ($t_t$) | −2.3311 | 0.4060 | −2.2961 | 0.4236 |
| | None (t) | 1.4204 | 0.958 | 1.6213 | 0.9716 |
| LNAGRIC_GDP | Intercept ($t_m$) | 2.3348 | 0.9999 | 0.4891 | 0.9835 |
| | Intercept and Trend ($t_t$) | −1.6469 | 0.7486 | −5.3864 | 0.0007 *** |
| | None (t) | 5.6052 | 1.000 | 2.4414 | 0.9954 |
| LNTRADE_OPEN | Intercept ($t_m$) | −0.7824 | 0.8102 | −0.6771 | 0.8381 |
| | Intercept and Trend ($t_t$) | −2.3159 | 0.4136 | −2.2807 | 0.4314 |
| | None (t) | 1.4278 | 0.9586 | 1.6278 | 0.9720 |
| At 1st difference—I(1) | | | | | |
| d(LNCO$_2$) | Intercept ($t_m$) | −6.2737 | 0.0000 *** | −6.2753 | 0.0000 *** |
| | Intercept and Trend ($t_t$) | −6.8874 | 0.0000 *** | −7.1838 | 0.0000 *** |
| | None (t) | −5.8419 | 0.0000 *** | −5.8284 | 0.0000 *** |
| d(LNRENP) | Intercept ($t_m$) | −6.5291 | 0.0000 *** | −6.5291 | 0.0000 *** |
| | Intercept and Trend ($t_t$) | −6.5218 | 0.0000 *** | −6.5332 | 0.0000 *** |
| | None (t) | −6.1792 | 0.0000 *** | −6.1562 | 0.0000 *** |
| d(LNAGRIC_GDP) | Intercept ($t_m$) | −13.7683 | 0.0000 *** | −10.0616 | 0.0000 *** |
| | Intercept and Trend ($t_t$) | −14.6062 | 0.0000 *** | −11.4813 | 0.0000 *** |
| | None (t) | 0.3301 | 0.0000 *** | −8.1738 | 0.0000 *** |
| d(LNTRADE_OPEN) | Intercept ($t_m$) | −6.5184 | 0.0000 *** | −6.5184 | 0.0000 *** |
| | Intercept and Trend ($t_t$) | −6.5135 | 0.0000 *** | −6.5244 | 0.0000 *** |
| | None (t) | −6.1665 | 0.0000 *** | −6.1434 | 0.0000 *** |

*** Denotes significance at 1% level.

**Table 5.** ARDL Bounds Test.

| | | | Critical Values | | | | | | |
|---|---|---|---|---|---|---|---|---|---|
| Lag Length | F-Statistic | k | 10% | | 5% | | 1% | | Outcome |
| | | | Lower Bound | Upper Bound | Lower Bound | Upper Bound | Lower Bound | Upper Bound | |
| ARDL(4,3,4,2) | 17.466 | 3 | 2.676 | 3.586 | 3.272 | 4.306 | 4.614 | 5.966 | |
| | | | | | | | | | Cointegrated |

In order to test for the existence of the long-run relationship, the cointegration is estimated using the ordinary least squares (OLS) method. The bound test null and alternative hypotheses are as follows:

$$H_0 = \varnothing_1 = \varnothing_2 = \varnothing_3 = 0;$$

$$H_1 \neq \varnothing_1 \neq \varnothing_2 \neq \varnothing_3 \neq 0.$$

The results in Table 5 indicate that the F statistics bound test value of 17.47 is higher than all the lower- and upper-bound values at 1%, 5% and 10% levels of statistical significance. We can then conclude that there is a long cointegration relationship in the series and therefore a long-run and short-run relationship can be estimated by the ARDL model.

### 5.5. Lag Selection

Various criteria such as LogL, LR, FPE, AIC, SC and HQ are used to determine optimal lags. The maximum lags were determined using the Akaike Information criteria (AIC), which are commonly used to determine the optimal lags for the ARDL model. Table 6 shows the unrestricted vector autoregressive (VAR) model lag length selection. However, all six criteria confirmed lag 1 to be best suited for the model.

**Table 6.** Lag length selection.

| Lag | LogL | LR | FPE | AIC | SC | HQ |
|-----|------|------|------|------|------|------|
| 0 | 232.230 | NA | 0.00 | −15.215 | −15.029 | −15.156 |
| 1 | 322.504 | 150.457 * | 0.00 * | −20.167 * | −19.233 * | −19.868 * |
| 2 | 337.651 | 21.206 | 0.00 | −20.11 | −18.429 | −19.572 |

* Indicates lag order selected by the criterion.

### 5.6. ARDL Error Correction and Long-Run Results

Short-run results in Table 7 show that all variables were statistically significant. Carbon emissions and trade openness had positive coefficient, except for the lags which were expected and statistically significant at a 1% level of significance. Agricultural GDP, renewable energy production, carbon emissions and trade openness converge in the long run at an equilibrium speed of −0.93 or at 93%, which is statistically significant at a 1% level of significance. Agricultural GDP will converge to its long-run equilibrium by 93% speed of adjustment alongside $CO_2$, renewable energy and trade openness activities in the economy. In period terms, it will take 1.08 years for all variables to converge into an equilibrium. The positive coefficient means that a 1% increase in carbon emissions and trade openness increases agricultural GDP by 0.25% and 70.16%, respectively. Renewable energy production had a negative coefficient but was statistically significant at a 1% level of significance. Considering that most countries, including South Africa, are just starting to build a conducive policy environment for renewable energy production, this is expected. The current results, however, suggest that the renewable energy generation regulatory environment slows down agricultural economic growth; thus, a 1% increase in renewable energy generation leads to a 69.86% decrease in agricultural GDP activities. The long-run relationship was further confirmed by the bound test results. In the long run, as indicated in Table 8, carbon emissions and trade openness were positively significant, meaning that a 1% increase in carbon emissions and trade openness increases agricultural GDP by 1.05% and 39.42%, respectively. All the variables in the model were well-fitted with an F-statistics value of 50.63. The model was an overall good fit, with an adjusted R-square of 0.96, implying that 96% of the variation in agricultural GDP was explained by carbon emissions, renewable energy production and trade openness in the model. The causality test in Table 9 indicates a bidirectional causality between renewable energy production and trade openness and therefore the null hypothesis is rejected at a 10% level of significance. This means that trade openness can predict renewable energy production and renewable energy production can also predict trade openness.

**Table 7.** Error correction estimates.

| Dependent Variable: D(LNAGRIC_GDP) | | | | |
|---|---|---|---|---|
| **Selected Model: ARDL(4,3,4,2)** | | | | |
| **Variable** | **Coefficient** | **Std. Error** | **t-Statistic** | **Prob.** |
| COINTEQ *(ECM) | $-0.928$ | 0.085 | $-10.913$ | 0.000 |
| D(LNAGRIC_GDP($-1$)) | 0.110 | 0.075 | 1.472 | 0.162 |
| D(LNAGRIC_GDP($-2$)) | 0.016 | 0.065 | 0.248 | 0.808 |
| D(LNAGRIC_GDP($-3$)) | 0.160 | 0.059 | 2.703 | 0.016 |
| D(LNRENP) | $-69.861$ | 12.676 | $-5.511$ | 0.000 |
| D(LNRENP($-1$)) | 113.350 | 13.248 | 8.556 | 0.000 |
| D(LNRENP($-2$)) | 0.079 | 0.035 | 2.256 | 0.039 |
| D(LNCO$_2$) | 0.254 | 0.102 | 2.496 | 0.025 |
| D(LNCO$_2$($-1$)) | $-0.950$ | 0.139 | $-6.826$ | 0.000 |
| D(LNCO$_2$($-2$)) | $-0.424$ | 0.123 | $-3.450$ | 0.004 |
| D(LNCO$_2$($-3$)) | $-1.371$ | 0.134 | $-10.256$ | 0.000 |
| D(LNTRADE_OPEN) | 70.157 | 12.669 | 5.538 | 0.000 |
| D(LNTRADE_OPEN($-1$)) | $-113.443$ | 13.246 | $-8.564$ | 0.000 |
| R-squared | 0.976 | Mean dependent var | | 0.027 |
| Adjusted R-squared | 0.957 | S.D. dependent var | | 0.087 |
| S.E. of regression | 0.018 | Akaike info criterion | | $-4.869$ |
| Sum squared resid | 0.005 | Schwarz criterion | | $-4.250$ |
| Log likelihood | 81.162 | Hannan−Quinn criteria. | | $-4.680$ |
| F-statistic | 50.631 | Durbin−Watson stat | | 2.837 |
| Prob(F-statistic) | 0.000 | | | |

* $p$-values are incompatible with t-bounds distribution.

**Table 8.** ARDL Long-run results.

| **Variable** | **Coefficient** | **Std. Error** | **t-Statistic** | **Prob.** |
|---|---|---|---|---|
| LNRENP($-1$) | $-39.239$ | 15.208 | $-2.580$ | 0.016 ** |
| LNCO$_2$($-1$) | 1.054 | 0.082 | 12.868 | 0.000 *** |
| LNTRADE_OPEN($-1$) | 39.418 | 15.113 | 2.608 | 0.015 ** |

** 5% and *** 1% level of statistical significance.

**Table 9.** Pairwise Granger Causality Tests.

| **Null Hypothesis:** | **F-Statistic** | **Prob.** |
|---|---|---|
| D(LNRENP) does not Granger Cause D(LNAGRIC_GDP) | 1.871 | 0.176 |
| D(LNAGRIC_GDP) does not Granger Cause D(LNRENP) | 1.095 | 0.351 |
| D(LNCO$_2$) does not Granger Cause D(LNAGRIC_GDP) | 0.24 | 0.789 |
| D(LNAGRIC_GDP) does not Granger Cause D(LNCO$_2$) | 0.015 | 0.985 |
| D(LNTRADE_OPEN) does not Granger Cause D(LNAGRIC_GDP) | 1.866 | 0.176 |
| D(LNAGRIC_GDP) does not Granger Cause D(LNTRADE_OPEN) | 1.105 | 0.348 |
| D(LNCO$_2$) does not Granger Cause D(LNRENP) | 0.429 | 0.656 |
| D(LNRENP) does not Granger Cause D(LNCO$_2$) | 0.22 | 0.805 |
| D(LNTRADE_OPEN) does not Granger Cause D(LNRENP) | 3.004 | 0.069 * |
| D(LNRENP) does not Granger Cause D(LNTRADE_OPEN) | 3.023 | 0.067 * |
| D(LNTRADE_OPEN) does not Granger Cause D(LNCO$_2$) | 0.22 | 0.804 |
| D(LNCO$_2$) does not Granger Cause D(LNTRADE_OPEN) | 0.432 | 0.654 |

* Denotes reject the null.

*5.7. Diagnostic Test*

Different diagnostic tests are normally used to assess the stability of the model. The paper used the three tests in Table 10 below and passed the linear regression assumptions. The results in Table 10 indicate that the model had normally distributed residuals and was free from serial correlation and heteroskedasticity. Figure 3 indicates the stability structure

of the model using cumulative sum (CUSUM) and cumulative sum of squares (CUSUMSQ). The middle lines indicate that coefficients are stable at a 5% level of significance.

**Table 10.** ARDL diagnostic test.

| Diagnostic Statistics | *p*-Values | Outcome |
| --- | --- | --- |
| Breusch–Godfrey LM | 0.260 | Serial correlation free |
| Breusch–Pagan–Godfrey | 0.906 | Heteroskedasticity free |
| Jarque–Bera Test | 0.511 | Normal residuals |

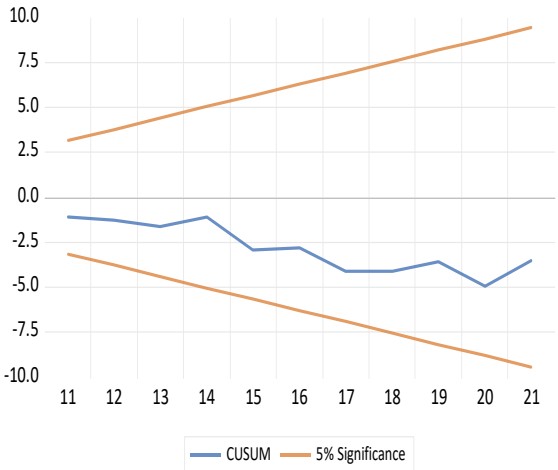 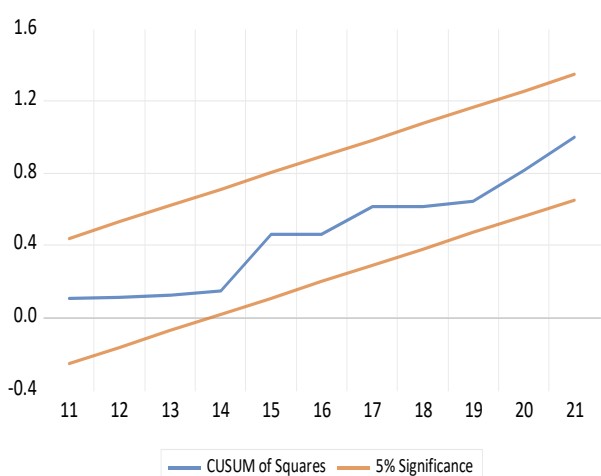

**Figure 3.** CUSUM and CUSUMSQ plots.

*5.8. Model Robustness*

In the long-run model, renewable energy generation, carbon emission and trade openness were all statistically significant. $CO_2$ and trade openness had positive co-efficient while renewable energy generation had a negative co-efficient. The FMOLS, DOLS and CCR model in Table 11 shows similar results as that of the long-run model, which is an expected sign of model robustness. In all three models, renewable energy generation is statistically significant with a negative coefficient, while $CO_2$ emissions and trade openness are statistically significant with a positive coefficient. In conclusion, the FMOLS, DOLS and CCR models confirmed the ARDL long-run model.

**Table 11.** Alternative results.

| FMOLS | | | | | DOLS | | | | | CCR | | | | |
| --- | --- | --- | --- | --- | --- | --- | --- | --- | --- | --- | --- | --- | --- | --- |
| Variable | Coefficient | Std. Error | t-Statistic | Prob. | Variable | Coefficient | Std. Error | t-Statistic | Prob. | Variable | Coefficient | Std. Error | t-Statistic | Prob. |
| LNRENP | −77.703 | 21.805 | −3.564 | 0.001 | LNRENP | −43.194 | 22.440 | −1.925 | 0.072 | LNRENP | −78.689 | 26.153 | −3.009 | 0.006 |
| LNTRADE_OPEN | 77.707 | 21.696 | 3.582 | 0.001 | LNTRADE_OPEN | 43.449 | 22.324 | 1.946 | 0.069 | LNTRADE_OPEN | 78.682 | 26.023 | 3.024 | 0.005 |
| LNCO201 | 1.061 | 0.167 | 6.354 | 0.000 | LNCO201 | 0.888 | 0.154 | 5.763 | 0.000 | LNCO201 | 1.071 | 0.186 | 5.774 | 0.000 |
| C | 15.309 | 0.891 | 17.182 | 0.000 | C | 16.022 | 0.809 | 19.815 | 0.000 | C | 15.266 | 0.981 | 15.566 | 0.000 |
| $R^2$ | 0.855 | | | | $R^2$ | 0.94545 | | | | $R^2$ | 0.854398 | | | |

## 6. Discussion

This paper's aim was to assess the relationship between agricultural economic growth and renewable energy supply, carbon emission and trade openness. A cointegration test was performed to ascertain a long-run cointegration and an error correction model was estimated for a short-run relationship. The bounds test revealed that a long-run relationship existed among the series. The result showed that an increase in carbon emissions and trade openness increases agricultural economic growth. The findings are in line with several studies: Ali et al. (2021) found that an increase in agricultural value added per capita increased carbon emissions; Ntim-Amo et al. (2021) found that agricultural economic

growth had significant positive effects on $CO_2$ emissions; Wang (2022) found that $CO_2$ emissions have significant effects on agricultural economic growth; Adebayo et al. (2021) found a positive correlation between agriculture, value added and $CO_2$ emissions; Phiri et al. (2021) also found a positive relationship between agriculture and $CO_2$ emissions; Sertoglu (2021) also found similar results; Ramzan et al. (2021) found that $CO_2$ emission could significantly predict agriculture productivity; Orhan et al. (2021) observed that agriculture is a crucial determinant of $CO_2$ emissions; Zaman et al. (2022) found that adding value in agricultural production helps reduce $CO_2$ emissions; Adedoyin et al. (2021) found value-added agriculture to be a driver of $CO_2$ emissions; Raihan and Tuspekova (2022b) observed that value added agriculture causes environmental degradation. Karimi Alavijeh et al. (2022) also observed that agricultural value added had a positive relationship with $CO_2$ emissions. In contrast, a good relationship between agriculture and environmental quality was observed by Lin et al. (2022), who observed a negative relationship between agricultural GDP and $CO_2$ emissions; Gurbuz et al. (2021) also found that agricultural value added had a negative impact on $CO_2$ emissions; Raihan and Tuspekova (2022a, 2022c, 2022d) found an increase in agricultural productivity leads to $CO_2$ emissions reduction; Selcuk et al. (2021) observed that agriculture has a significantly negative association with $CO_2$; Raihan et al. (2022) found reduced agricultural productivity increase $CO_2$ emissions; Zafar et al. (2022) observed that agricultural development improves environmental quality; a study by Udemba et al. (2022) also showed a negatively significant relationship between the carbon emission and agriculture.

Results showed a positive relationship between agricultural economic growth and trade openness, which was expected. Various studies found a positive relationship between trade openness and economic growth, such as a study by Ibrahim et al. (2022), which found that trade openness significantly improved agricultural sector performance, as well as studies by Siregar and Widjanarko (2022), Ashraf et al. (2022), Nguyen (2022), Chowdhary and Joshi (2022), Islam et al. (2022), Qi et al. (2022), Jirbo et al. (2022) and Dahmani et al. (2022). However, Rasoanomenjanahary et al. (2022) found an inverse relationship between trade openness and economic growth.

The result also showed an inverse relationship between agricultural economic growth and renewable energy supply. The findings suggest that renewable energy supply slows down the agricultural sector's economic growth. The results were expected, considering that South Africa's sector heavily depends on fossil fuel (more than 70%), and the adoption of renewable energy is low, a common reality in developing countries. The negative relationship suggests that renewable energy is currently expensive for the agricultural sector and the demand will be low. This negative relationship is in line with several studies in developing countries by Ocal and Aslan (2013), Maji et al. (2019), Akram et al. (2021), Fotio et al. (2022) and Wang et al. (2022). Positive results between renewable energy and agricultural growth were found in other studies by Chopra et al. (2022), Chen et al. (2022a) and Magazzino et al. (2022).

## 7. Conclusions and Policy Recommendations

An ARDL model was used to estimate short-run and long-run relationships based on cointegration results between agricultural economic growth and renewable energy supply, carbon emission and trade openness. A study of this nature is important as the agricultural sector is an important component of the economy, providing employment for the most vulnerable communities in South Africa. Moreover, in addition to traditional factors of production such as labour, capital, entrepreneurship, technology and land, energy has become a distinct official factor of production in growing the economy in the context of climate change in recent years. On the other hand, environmental economics studies have confirmed that a relationship exists between economic growth, and carbon emissions from the Environmental Kuznets Curve theory (Fritz and Koch 2016). Other studies have further linked this relationship with energy supply and consumption. Strong evidence shows that the negative effects of climate change affect economic growth adversely. Considering that

agricultural activities and energy production and consumption account for most carbon emissions, the current policy advocacy directs economies to be fuelled by renewable energy to mitigate climate change's negative effects. However, owing to the limited scale of renewable energy supply worldwide, and more so in developing countries, a negative relationship has been observed between economic growth and renewable energy supply and consumption. In this pretext, it was therefore necessary to examine if a similar nexus exists between agricultural economic growth, renewable energy supply and carbon emissions, with trade openness as a control parameter. This paper has applied the error correction model to assess the nexus between agricultural GDP, renewable energy production, carbon dioxide emissions and trade openness in South Africa. The estimation results showed a positive relationship between agricultural economic growth and carbon emissions and trade openness, implying that an increase by a unit percentage in the two variables will increase agricultural economic growth. This positive relationship was, however, expected a priori.

The results confirm that economic growth deteriorates environmental quality. A priori, growth in the agricultural economy was expected to increase carbon emissions, as informed by the Environmental Kuznets Curve hypothesis. The findings in the study were consistent with those of various authors (Rehman et al. 2019; Aydoğan and Vardar 2020; Eyuboglu and Uzar 2020; Yurtkuran 2021; Usman et al. 2021; Yasmeen et al. 2022).

Results also showed that international trade increased agricultural economic growth. This was expected, as open markets enable competitiveness in the economy, ultimately creating more opportunities for global trade. Trade openness benefits economic growth (Qi et al. 2022; Gniniguè et al. 2022; Pellegrina 2022). Farrokhi and Pellegrina (2021) also established that trade improves agricultural productivity activities.

However, a negative relationship was observed between renewable energy production and agricultural economic growth, also confirming what has been observed in other empirical studies. The results showed that agricultural economic growth will increase in the short and long run when carbon emission and trade openness increase. However, weaker agricultural economic growth was observed as the renewable energy supply increased. Looking at the low scale of renewable energy investments in the country, and in line with some authors' findings (Akram et al. 2021; Fotio et al. 2022), the expectation was that renewable energy production would not play a significant role in the agricultural economy. Thus, renewable energy supply reduction accelerates agricultural economic growth. The reality is that substantial renewable energy investments have not been made in developing economies, especially in Africa, precisely because most countries do not have the budget for renewable energy production. This is the reason why during COP26 (2021) in Glasgow, African leaders indicated the need for developed countries to fund clean energy transition projects. During COP27, there was a sense that developed countries should not be tempted to loan climate finance to developing countries but fund them instead. The observed negative relationship between agricultural economic growth and renewable energy was expected, as renewable energy in South Africa is still expensive and consequently has a negative impact on agricultural economic growth. Additionally, the goal of this research is to fill a major knowledge gap, notably in South Africa, in the discourse of renewable energy in the agricultural sector and to assist policymakers in formulating plans to achieve carbon neutrality in this sector.

The study, therefore, makes recommendations for developing renewable energy to lessen reliance on the consumption of fossil fuels. Taking note of other forms of green energy sources, such as solar, wind and hydro energy, the study focuses on bioenergy as the farming sector is directly involved and less attention and investment are given to bioenergy. One of the major obstacles to renewable energy in the agricultural sector in South Africa is the lack of a conducive legislative and policy environment specifically targeted at the agricultural sector. When there is no proper policy in place addressing how feedstock from farmers can be harnessed and incentivized, this will derail green energy transition progress, regardless of the magnitude of investment committed. The policy

should include investments and that is a gap that is currently existing in South Africa in the agricultural sector. To comprehensively provide a conducive policy environment, (a) the study recommends special Research and Development (R&D) finance in renewable energy, specifically in the agricultural sector, in order to unpack all the socioeconomic aspects of bioenergy development. Currently, such funds are dedicated to solar and wind. A portion of this fund can be funded by the current fuel levy taxation system. (b) Based on R&D outcomes, the study recommends a concerted effort to develop renewable energy vocational programmes at institutions of higher learning in South Africa. At present, a huge gap in renewable energy human development exists in the country. Of the 26 universities in South Africa, no university is currently offering a certificate, diploma or degree in green energy-related aspects. The country needs technicians, professionals and researchers in this field in order to build infrastructure and vocational skills in national, regional and rural areas for sustainability. (c) The study recommends a renewable energy policy for the agricultural sector to encourage public and private sector involvement with tax cut bonuses, an incentive that has been suggested by various scholars. (d) Considering that the agricultural sector provides the feedstock for bioenergy energy production—that is, biogas (livestock and other residues), biofuel and biodiesel (oil crops etc.) and co-generation from biomass (sugarcane, tobacco, industrial hemp (cannabis), sorghum, forestry, maize residues, indigenous crops etc.)—there is a need to create an incentive scheme for renewable energy production in the agricultural sector. This will encourage farmers to participate as suppliers and consumers of renewable energy. With the just clean energy transition (loosely translated 'leave no one behind') debate on the table in the country, the approach should be directed at the smallholder and emerging farmers in the agricultural sector, coupled with clean energy finance. Finally, (e) the study recommends that government must create tariffs that renewable energy producers in the agricultural sector can use to sell excess energy to the national grid, a common practice by countries who have made strides in renewable energy production at the rural level. This will clarify the issues of access of the small-scale agricultural sector to the renewable energy market in the country, since most fear that the green economy will mostly exclude small-scale farmers and include commercial farmers. This approach will encourage the participation of the agricultural sector at both the small-scale and at the commercial level, and also encourage private investments in renewable energy in the agricultural sector. The main benefit is quality of life in the long run as carbon emissions will be reduced.

*Future Areas of Research*

It is important to note that the study only took carbon emission, renewable energy production and openness to trade into account in its assessment of agricultural economic growth. The results pertain to the readiness of green energy transition in the country and how this affects the agricultural industry. The study, however, is limited, and additional macroeconomic variables can be added to expand the work. In addition, future studies can use other proxies for environmental quality, such as ecological footprint, and trade openness proxies, such as composite trade intensity (CTI), green technology investment, ICT, government expenditure on agriculture and renewable energy consumption.

**Funding:** This research received no external funding.

**Informed Consent Statement:** Not applicable.

**Data Availability Statement:** The dataset is available upon reasonable request.

**Conflicts of Interest:** Authors declare no conflicts of interest.

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
