# Peer review of "Agricultural Economic Growth, Renewable Energy Supply and CO2 Emissions Nexus"

_economies, doi:10.3390/economies11030085_

Round 1
Reviewer 1 Report
These are the most important comments that I consider necessary to be incorporated in the document:
1. I recommend that the Abstract should be slightly adjusted. Also, I recommend that it should be corrected to provide more fluency in English.
Although they present the aim, they should however present the usefulness of the paper. The method should be briefly described without getting into too much technical details. In addition, the authors mention “The study makes a new contribution in providing empirical evidence of the links between renewable energy supply and agricultural GDP which can drive policy on renewable energy use in the agricultural sector.”. There is nothing new from my understanding given that the authors, at the end of the paper, cite some works that deal with the same nexus. And I recommend that, instead of providing the statistical interpretation of the model, they should present the socio-economic implications of these results (The results showed a positive relationship between agricultural economic growth, carbon dioxide and trade openness in the short and long run. An inverse relationship between renewable energy production and agricultural economic growth was observed.)
2. With regard to the Introduction section: I appreciate the authors describing the context of the research. But they stop here. I believe that it should be followed by: a clear and straight forward aim of the paper, research questions, hypotheses, brief description of the methodology and novelty of the paper. How is this research contributing to the existing strand of literature? What exactly you do better than the existing works? And from my understanding from the end of the paper, there are many who study the same thing.
3. With regard to the Literature review section, I did not notice a logic in enumerating the existing works. There are two paragraphs dedicated to the literature, but I did not saw a noticeable distinction between the two groups of literature descriptions. I highly recommend to focus on explaining what previous studies have done and what they have not done, and how your study is different in this respect. Studying the literature should not consist only in listing the papers with just a rudimentary idea.
For example:
[A study by Banks and Schäffler (2005) concluded that increased usage of renewable energy would also lessen South Africa's reliance on the fluctuating (and rising) costs of imported fuels, Aliyu et al. (2018) reviewed renewable energy development in Africa with a focus in South Africa, Egypt and Nigeria, the review suggested proper technology, awareness and skills attention for renewable energy production.]. COULD BE FOLLOWED BY {In this respect we believe that……}
[Ibrahim et al. (2021) reviewed African renewable energy production using Nigeria, Cameroon, Ghana and South Africa as a case study, The study found that subsidies for the tax on renewable energy can encourage energy production]. COULD BE FOLLOWED BY {We also agree with this, but from our point of view….} {With respect to this authors, our opinion is that it should/could/might ….} {This paper unfortunately does not consider …}
4. I recommend moving Table 1-3 in the Annex. These are too basic.
5. I recommend looking carefully again at the results in your tables. Something seems abnormal. For example, if I’m not mistaking, in table 7, your variables explain around 95% (adjusted R) of the movement of your main variable (which seems a lot in such models). Do you have similar studies to cite with the same values? How do you explain that your variables with lag 0 have a sign, and with lag 1 have a different sign? Don’t you find unusual that in table 3 you have a correlation coefficient of 0.999. That is almost impossible when you are dealing with financial data. And does it not affect the above mentioned value of 95%? (autocorrelation?)
6. Within lines 318 and 338 you provide a cascade of works where you constantly shift between: “agricultural value added had significant positive effects on CO2 emissions” and “CO2 emissions have significant effects on agricultural economic growth”. You should be more consistent and better structure how you present other works.
7. I recommend that the author improves the theoretical foundations upon which the analysis is built. There is too much statistical interpretation and less socio-economic meaning. It lacks the story. I understand the model but I believe that it lacks the story behind it. They need to give a theoretical argument to this by devoting a proposition and a theoretical framework accompanying it. Hence, I would suggest to focus on the theoretical mechanisms. You should also build on these the hypotheses.
8. There are some mistakes and oversights existing in the text (and English misspellings).
9. With regard to the Reference section, for a paper that wants to be published in “Economies”, I am sure that the authors have previously checked the other papers published in this journal and could refer to some of these.
Author Response
Thank you very much for the invaluable comments you have made to improve the work.
Response to comments
Reviewer 1 (All Responses highlighted in green in the document)
|
Section & page |
comment |
Response |
|
Abstract |
1. I recommend that the Abstract should be slightly adjusted. Also, I recommend that it should be corrected to provide more fluency in English. Although they present the aim, they should however present the usefulness of the paper.
The method should be briefly described without getting into too much technical details.
In addition, the authors mention “The study makes a new contribution in providing empirical evidence of the links between renewable energy supply and agricultural GDP which can drive policy on renewable energy use in the agricultural sector.” There is nothing new from my understanding given that the authors, at the end of the paper, cite some works that deal with the same nexus.
And I recommend that, instead of providing the statistical interpretation of the model, they should present the socio-economic implications of these results (The results showed a positive relationship between agricultural economic growth, carbon dioxide and trade openness in the short and long run. An inverse relationship between renewable energy production and agricultural economic growth was observed.)
|
The usefulness of the paper was added to the abstract (see more information added from lines 14-15.
Language editing is to be done once all comments from reviewers are incorporated.
The abstract briefly described the ARDL method employed in the study.
It is true that a study of this nature is not new globally but new in South Africa, the cited work emanates from other countries. The thinking behind this is that, countries differ and as such results will carry different implications. ‘in South Africa’ was added to the statement to provide more clarity.
The socio-economic implications were added to improve the interpretation of the model (see more highlighted information added).
|
|
Introduction |
2. With regard to the Introduction section: I appreciate the authors describing the context of the research. But they stop here. I believe that it should be followed by: a clear and straight forward aim of the paper, research questions, hypotheses, brief description of the methodology and novelty of the paper. How is this research contributing to the existing strand of literature? What exactly you do better than the existing works? And from my understanding from the end of the paper, there are many who study the same thing. |
The research aim, questions and hypothesis were added to improve the work. The novelty and the contribution of the paper were added and clearly defined. (see more highlighted information added).
|
|
Literature review |
3. With regard to the Literature review section, I did not notice a logic in enumerating the existing works. There are two paragraphs dedicated to the literature, but I did not saw a noticeable distinction between the two groups of literature descriptions. I highly recommend to focus on explaining what previous studies have done and what they have not done, and how your study is different in this respect. Studying the literature should not consist only in listing the papers with just a rudimentary idea.
For example: A study by Banks and Schäffler (2005) concluded that increased usage of renewable energy would also lessen South Africa's reliance on the fluctuating (and rising) costs of imported fuels, Aliyu et al. (2018) reviewed renewable energy development in Africa with a focus in South Africa, Egypt and Nigeria, the review suggested proper technology, awareness and skills attention for renewable energy production.].COULD BE FOLLOWED BY {In this respect we believe that……} |
The literature review section has been modified accordingly. The sections now explain the gap that the study intends to close. The voice of the author has been added to the review as suggested (see more highlighted information added under the literature section).
|
|
|
4. I recommend moving Table 1-3 in the Annex. These are too basic. |
Although the tables are basic, the author thought these tables were important especially because it is a conventional way of writing and provides authors with a step-by-step reference guide. |
|
|
5. I recommend looking carefully again at the results in your tables. Something seems abnormal. For example, if I’m not mistaking, in table 7, your variables explain around 95% (adjusted R) of the movement of your main variable (which seems a lot in such models). Do you have similar studies to cite with the same values? How do you explain that your variables with lag 0 have a sign, and with lag 1 have a different sign? Don’t you find unusual that in table 3 you have a correlation coefficient of 0.999. That is almost impossible when you are dealing with financial data. And does it not affect the above-mentioned value of 95%? (autocorrelation?) |
i) Time series data usually has a high Adjusted R. So this is very normal.
To mention a few, see similar studies : 1. Chen, Y., Wang, Z. and Zhong, Z., 2019. CO2 emissions, economic growth, renewable and non-renewable energy production and foreign trade in China. Renewable energy, 131, pp.208-216. 2. Liu, X., Zhang, S. and Bae, J., 2017. The nexus of renewable energy-agriculture-environment in BRICS. Applied energy, 204, pp.489-496. 3. Nguyen, H.T., Van Nguyen, S., Dau, V.H., Le, A.T.H., Nguyen, K.V., Nguyen, D.P., Bui, X.T. and Bui, H.M., 2022. The nexus between greenhouse gases, economic growth, energy and trade openness in Vietnam. Environmental Technology & Innovation, 28, p.102912. 4. Dogan, N., 2016. Agriculture and Environmental Kuznets Curves in the case of Turkey: evidence from the ARDL and bounds test. Agricultural Economics, 62(12), pp.566-574. 5. Hadi, S.N. and Chung, R.H., 2022. Estimation of Demand for Beef Imports in Indonesia: An Autoregressive Distributed Lag (ARDL) Approach. Agriculture, 12(8), p.1212. 6. Akram, R., Chen, F., Khalid, F., Huang, G. and Irfan, M., 2021. Heterogeneous effects of energy efficiency and renewable energy on economic growth of BRICS countries: a fixed effect panel quantile regression analysis. Energy, 215, p.119019. 7. Fotio, H.K., Poumie, B., Baida, L.A., Nguena, C.L. and Adams, S., 2022. A new look at the growth-renewable energy nexus: Evidence from a sectoral analysis in Sub-Saharan Africa. Structural Change and Economic Dynamics, 62, pp.61-71.
ii) Strong correlation amongst variables is not unusual but very common, it depends on the nature of a variable. (See Tabash, M.I., Farooq, U., Safi, S.K., Shafiq, M.N. and Drachal, K., 2022. Nexus between Macroeconomic Factors and Economic Growth in Palestine: An Autoregressive Distributed Lag Approach. Economies, 10(6), p.145.) iii) Autocorrelation was thoroughly checked and the diagnostic test in Table 10 confirms that the Durbin-Watson stats. also confirmed a negative correlation.
iii) lagged variables with a positive and a negative sign: Authors may report only the lagged value of interest, however, many show all the lags and their values, see examples in papers: 1. Tabash, M.I., Farooq, U., Safi, S.K., Shafiq, M.N. and Drachal, K., 2022. Nexus between Macroeconomic Factors and Economic Growth in Palestine: An Autoregressive Distributed Lag Approach. Economies, 10(6), p.145.) 2. Phiri, J., Malec, K., Kapuka, A., Maitah, M., Appiah-Kubi, S.N.K., Gebeltová, Z., Bowa, M. and Maitah, K., 2021. Impact of Agriculture and Energy on CO2 Emissions in Zambia. Energies, 14(24), p.8339. 3. Haider, A., Rankaduwa, W., ul Husnain, M.I. and Shaheen, F., 2022. Nexus between agricultural land use, economic growth and N2O emissions in Canada: is there an environmental Kuznets curve?. Sustainability, 14(14), p.8806. |
|
|
6. Within lines 318 and 338 you provide a cascade of works where you constantly shift between: “agricultural value added had significant positive effects on CO2 emissions” and “CO2 emissions have significant effects on agricultural economic growth”. You should be more consistent and better structure how you present other works. |
Corrected on page 13 (see highlighted in green). |
|
|
7. I recommend that the author improves the theoretical foundations upon which the analysis is built. There is too much statistical interpretation and less socio-economic meaning. It lacks the story. I understand the model but I believe that it lacks the story behind it. They need to give a theoretical argument to this by devoting a proposition and a theoretical framework accompanying it. Hence, I would suggest to focus on the theoretical mechanisms. You should also build on these the hypotheses. |
A new theoretical framework section was added on page 4. The section provides the story and the argument behind the model in the study. |
|
|
9. With regard to the Reference section, for a paper that wants to be published in “Economies”, I am sure that the authors have previously checked the other papers published in this journal and could refer to some of these. |
10 relevant reference sources from Economies ‘work were added to improve the work. |

Reviewer 2 Report
The authors proposed an interesting study "Agricultural economic growth, renewable energy supply and Co2 emissions nexus: Using ARDL approach". If authors are willing to incorporate following suggestions, then I would be willing to reconsider my decision.
1. Please spell out acronyms on the first mention. Even if you have defined an acronym in the abstract, it has to be defined again in the paper when first mentioned. Moreover, no synonym should be used in abstract.
2. The authors should revise the keywords. Your research theme is agricultural economic growth yet it unfit to be mentioned as keyword? Why is that?
3. The major defect of this study is the debate or argument is not clearly stated in the introduction session. Hence, the contribution is weak in this manuscript. I would suggest the author enhance your theoretical discussion and arrives at your theoretical argument.
4. Only investigating the relationship of different variables is not a significant contribution to the existing literature. A volume of research is available on this issue and in my opening, this study adds very little to the available literature.
5. The authors should create different sub-sections under the "Literature review" heading to methodologically explain the findings from recent literature.
6. Introduction, literature review, empirical results, and discussion section should be critically evaluated by the authors. I recommend improving this section by critically analyzing the previous studies and arrive at their own argument. Following papers are recommended to use while expanding and improving this section of your paper.
https://link.springer.com/article/10.1007/s11356-022-23656-8
https://www.sciencedirect.com/science/article/pii/S0960148122012186
https://link.springer.com/article/10.1007/s11356-022-20782-1
https://www.sciencedirect.com/science/article/pii/S0960148122002075
https://www.sciencedirect.com/science/article/pii/S2772427122000304
7. The captions of many figures are sufficient. It must be further outlined as Figure 2A, 2B etc.
8. For empirical analysis the authors haven’t selected other empirical approaches i.e., AMG or CCEMG tests that have the better ability to tackle the cross-sectional issue.
9. Explanation of empirical findings can be better supplemented by the researchers by citing and explaining recent literature.
10. Also, I'm so disappointed regarding the policy formulation. It's very common and presented in many previous studies. I suggest to the authors that they should suggest some new practical and managerial implications for sustainable development.
11. What are the future research directions of this study?
Author Response
Thank you very much for the invaluable comments made to improve the work.
Response to comments
Reviewer 2 (All Responses highlighted in blue in the document)
|
Section & page |
comment |
Response |
|
Abstract |
1. Please spell out acronyms on the first mention. Even if you have defined an acronym in the abstract, it has to be defined again in the paper when first mentioned. Moreover, no synonym should be used in abstract. |
The acronyms were fully defined in the abstract as suggested. |
|
Keywords |
2. The authors should revise the keywords. Your research theme is agricultural economic growth yet it unfit to be mentioned as keyword? Why is that? |
The agricultural economic growth keyword was added. |
|
Introduction |
3. The major defect of this study is the debate or argument is not clearly stated in the introduction session. Hence, the contribution is weak in this manuscript. I would suggest the author enhance your theoretical discussion and arrives at your theoretical argument. |
The reviewer’s suggestion is noted, and the introduction section was improved from pages 2 to 3.
In addition, a new section on the theoretical framework was added to strengthen the argument on page 4. |
|
Variables in the study. |
4. Only investigating the relationship of different variables is not a significant contribution to the existing literature. A volume of research is available on this issue and in my opening, this study adds very little to the available literature. |
The reviewer’s comment is noted. However, it is noted that more variables could have been added and that was the initial intention. However, the model was constrained due to the limited number of observations and the depletion of degrees of freedom problem.
In hindsight, South Africa is currently in the debate of a just clean energy transition agenda, which basically means leaving no one behind in the green energy economy. The majority of the poor in South Africa are employed in the agricultural sector. Moreover, the majority of farmers in South Africa are small-scale farmers who are in rural areas. And the agricultural sector is the main feedstock producer for bioenergy, an area that has not been explored in South Africa. The current focus is mainly on solar and wind. And this is a strategic sector for achieving the just energy transition agenda.
This study brings a general understanding of the agricultural sector, renewable energy supply, and the environment and it is necessary and crucial at this stage for South Africa. It specifically gives everyone (policymakers, private & public institutions, academics etc.) a picture of the impact of renewable energy supply and where initial interventions should go to. Hence the recommendations which might sound elementary but very necessary and relevant on the ground were made in this study. The recommendations were elementary, however, experience working on the ground with rural farming communities shaped these sentiments.
It is true that similar studies have been conducted across the globe. However, even if that is the case, literature has shown different outcomes for various countries. An indication that there is a need to conduct individual studies.
From the studies reviewed, a few studies were observed, a study conducted in China (Chen, 2019) used renewable energy production to estimate economic growth. Most of the studies used consumption and various variables. The study does contribute to the existing literature. It is true that more variables could have been added but that leaves room for more research to be done to further investigate other potential predictors. |
|
Literature |
5. The authors should create different sub-sections under the "Literature review" heading to methodologically explain the findings from recent literature. |
The literature review was divided into sections and improved accordingly. |
|
Recommended papers |
6. Introduction, literature review, empirical results, and discussion section should be critically evaluated by the authors. I recommend improving this section by critically analyzing the previous studies and arrive at their own argument. Following papers are recommended to use while expanding and improving this section of your paper.
https://link.springer.com/article/10.1007/s11356-022-23656-8 https://www.sciencedirect.com/science/article/pii/S0960148122012186 https://link.springer.com/article/10.1007/s11356-022-20782-1 https://www.sciencedirect.com/science/article/pii/S0960148122002075 https://www.sciencedirect.com/science/article/pii/S2772427122000304
|
All the suggested papers were incorporated to improve the work on pages 5 and 6 |
|
|
7. The captions of many figures are sufficient. It must be further outlined as Figure 2A, 2B etc. |
Figures are labelled as suggested. |
|
|
8. For empirical analysis the authors haven’t selected other empirical approaches i.e., AMG or CCEMG tests that have the better ability to tackle the cross-sectional issue.
|
The reviewer's comment is noted, however, other studies have indicated that FMOLS, DOLS and CCR are sufficient to measure the robustness and that is why these methods were chosen. However, the suggested robust techniques are welcomed and will be used in future. The last paper suggested by the reviewer under recommended papers is one of the research work in support of these robust methods (See 1.Raihan, A. and Tuspekova, A., 2022. The nexus between economic growth, renewable energy use, agricultural land expansion, and carbon emissions: New insights from Peru. Energy Nexus, 6, p.100067. 2Sowah, J.K. and Kirikkaleli, D., 2022. Investigating factors affecting global environmental sustainability: evidence from nonlinear ARDL bounds test. Environmental Science and Pollution Research, 29(53), pp.80502-80519.) |
|
|
9. Explanation of empirical findings can be better supplemented by the researchers by citing and explaining recent literature. |
The findings were supported in the discussion section accordingly. |
|
|
10. Also, I'm so disappointed regarding the policy formulation. It's very common and presented in many previous studies. I suggest to the authors that they should suggest some new practical and managerial implications for sustainable development. |
The reviewer’s comments are well noted. The recommendation section was improved accordingly.
However, it is important to note that, the recommendations were probably elementary because they were also based on what has been observed on the ground working with farming communities in rural areas and government officials in other avenues and the lens of the recommendations will introduce basic ideas that might seem unimportant. The reality is that the agricultural sector can play a major role in the production of renewable energy, specifically bioenergy. Farmers already have the feedstock needed, however, government officials, extension officers, and farmers lack this knowledge. On the skills side, there are few to no people with green technology skills, hence the focus on the development of vocational programs. There is currently no green energy incentive for feedstock producers, an opportunity that can be harnessed by policymakers to encourage production. This is a necessary conversation that should be started so that in future this can find a place in policy. It is true that these recommendations might not be enough but key and practical in the current context. With an acknowledgement that this is not perfect, however, working with farmers on the ground, this seems to be relevant as well. |
|
|
11. What are the future research directions of this study? |
A future area of research was added accordingly. |

Reviewer 3 Report
The manuscript draft is devoted to an interesting problem that touches on analyzing the relationship between agricultural economic growth and renewable energy supply, carbon emission, and trade openness. The authors build an ARDL model that was used to estimate short-run and long-run relationships based on cointegration results. The experimental section is good. The proposed approach is logical, results are clear. However, I have the following remarks:
1. The title is too wide. The authors should detail the scope of the study in title.
2. The abstract does not provide clear information about the paper. The opening of the abstract is very far from the topic of the research. The authors should clarify the general topic under study and the central questions or statements of the problem their research addresses.
3. Introduction section is good. In the final part of the introduction, a brief overview of the rest of the paper should be written. It is appropriate for the authors to explain not only the structure but also the logic of the remainder of the paper.
4. The reference needs improvement. The geography of cited publications should also be widened. For example, the references could include the next papers:
Pata, U. K. (2021). Linking renewable energy, globalization, agriculture, CO2 emissions and ecological footprint in BRIC countries: A sustainability perspective. Renewable Energy, 173, 197-208.
Lamb, W. F., Wiedmann, T., Pongratz, J., Andrew, R., Crippa, M., Olivier, J. G., ... & Minx, J. (2021). A review of trends and drivers of greenhouse gas emissions by sector from 1990 to 2018. Environmental research letters, 16(7), 073005.
Klymenko, N., & Nehrey, M. (2022). Electricity Tariff Structures Modeling for Reengineering Ukrainian Energy Sector. In The International Conference on Artificial Intelligence and Logistics Engineering (pp. 493-502). Springer, Cham. DOI: 10.1007/978-3-031-04809-8_45
Duque-Acevedo, M., Belmonte-Ureña, L. J., Plaza-Úbeda, J. A., & Camacho-Ferre, F. (2020). The management of agricultural waste biomass in the framework of circular economy and bioeconomy: An opportunity for greenhouse agriculture in Southeast Spain. Agronomy, 10(4), 489.
5. The authors wrote section Results two times. It is appropriate for the authors to write the Methodology section. The authors should list potential weaknesses in methodology and present evidence supporting their choice.
6. The conclusion is not explained properly. The conclusion section should be extended using: a discussion of related research, and a comparison between the authors’ results and initial hypothesis.

Author Response
Thank you very much for the invaluable comments made to improve the work.
Response to comments
Reviewer 3 (All Response highlighted in Red in the document)
|
Section & page |
comment |
Response |
|
General |
The manuscript draft is devoted to an interesting problem that touches on analyzing the relationship between agricultural economic growth and renewable energy supply, carbon emission, and trade openness. The authors build an ARDL model that was used to estimate short-run and long-run relationships based on cointegration results. The experimental section is good. The proposed approach is logical, results are clear. |
Well noted. |
|
Title |
1. The title is too wide. The authors should detail the scope of the study in title. |
Done: The title was shortened to Agricultural economic growth, renewable energy supply, and Co2 emissions nexus. |
|
Abstract |
2. The abstract does not provide clear information about the paper. The opening of the |
Done: The abstract was improved by adding the general topic of the study and by also adding the research question (see highlighted info in red). |
|
Introduction |
3. Introduction section is good. In the final part of the introduction, a brief overview of |
Done: A brief structure of the paper was added (see highlighted info in red) providing logic for the remainder of the paper. |
|
Literature review |
4. The reference needs improvement. The geography of cited publications should also be Lamb, W. F., Wiedmann, T., Pongratz, J., Andrew, R., Crippa, M., Olivier, J. G., ... &
Klymenko, N., & Nehrey, M. (2022). Electricity Tariff Structures Modeling for Reengineering Ukrainian Energy Sector. In The International Conference on Artificial Duque-Acevedo, M., Belmonte-Ureña, L. J., Plaza-Úbeda, J. A., & Camacho-Ferre, |
Done: The suggested papers were used to strengthen the literature. Pata (more information added). Lamb et al (more information added) . Klymenko & Nehrey- this paper was not used as it was irrelevant to the study. It turned out to be focusing on technical aspects. Duque-Acevedo (more information added). |
|
Methodology |
5. The authors wrote section Results two times. It is appropriate for the authors to write |
Done: Section 3 was the methodology section; however, it was wrongly specified as Results. This has been corrected. The potential disadvantage was listed (more information was added and highlighted in red under this section). The choice of the model was explained and supported by work from five authors (see more information added highlighted in red). |
|
Conclusion |
6. The conclusion is not explained properly. The conclusion section should be extended
|
Done: The conclusion was extended; a priori expectations of the study were discussed. The study findings were compared with the findings of other authors (see more information added highlighted in red). |

Round 2
Reviewer 1 Report
I appreciate the effort made by the author to tackle all my suggestions. I believe that this version of the paper is considerably improved. Overall, the paper is ready to be published.
However, I would like to highlight the following: check all your references again! It is very important to have your references updated and correctly written (there are some in capital letters). (ex: Jirbo, B.V., Danladi, J. and Atayi, A.V.,) There is an additional space after Phillips, P.C. and Perron, P., 1988. I recommend using DOI.
Author Response
Thank you so much for all your efforts to improve the paper.
Response to comments
Reviewer 1 (All additions in track changes)
|
comment |
Response |
|
I appreciate the effort made by the author to tackle all my suggestions. I believe that this version of the paper is considerably improved. Overall, the paper is ready to be published. However, I would like to highlight the following: check all your references again! It is very important to have your references updated and correctly written (there are some in capital letters). (ex: Jirbo, B.V., Danladi, J. and Atayi, A.V.,)
There is an additional space after Phillips, P.C. and Perron, P., 1988. I recommend using DOI.
|
Well noted. Thank you.
DOI were added to references where available. |

Reviewer 2 Report
I am satisfied with the revision and receommend publication.
Author Response
Thank you so much for all your efforts to improve the paper.
Reviewer 3 Report
The paper can be published in present form
Author Response

(The authors gave the same response as above.)
